# The Impact of COVID-19 Pandemic on the Clinical Practice Patterns in Atrial Fibrillation: A Multicenter Clinician Survey in China

**DOI:** 10.3390/jcm11216469

**Published:** 2022-10-31

**Authors:** Feng Hu, Minhua Zang, Lihui Zheng, Wensheng Chen, Jinrui Guo, Zhongpeng Du, Erpeng Liang, Lishui Shen, Xiaofeng Hu, Xuelian Xu, Gaifeng Hu, Aihua Li, Jianfeng Huang, Yan Yao, Jun Pu

**Affiliations:** 1Department of Cardiology, Renji Hospital, School of Medicine, Shanghai Jiaotong University, Shanghai 200127, China; 2Department of Cardiology, Fuwai Hospital, National Center for Cardiovascular Diseases, Chinese Academy of Medical Sciences and Peking Union Medical College, Beijing 100037, China; 3Department of Cardiology, Guangdong Provincial Hospital of Chinese Medicine, Guangzhou 510120, China; 4Department of Cardiology, Fuwai Yunnan Cardiovascular Hospital, Kunming 650102, China; 5Department of Cardiology, Zhu Jiang Hospital of Southern Medical University, Guangzhou 510280, China; 6Heart Center of Henan Provincial People’s Hospital, Central China Fuwai Hospital, Zhengzhou University, Zhengzhou 451460, China; 7Department of Cardiology, Affiliated Hangzhou First People’s Hospital, Zhejiang University School of Medicine, Hangzhou 310003, China; 8Department of Cardiology, Shanghai Chest Hospital, Shanghai Jiao Tong University, Shanghai 200030, China; 9Department of Cardiology, University-Town Hospital of Chongqing Medical University, Chongqing 400042, China; 10Department of Cardiology, The First Affiliated Hospital of Wenzhou Medical University, Wenzhou 325035, China; 11Department of Cardiology, The Affiliated Hospital of Yangzhou University, Yangzhou 225007, China

**Keywords:** COVID-19 pandemic, atrial fibrillation, interventional therapy, catheter ablation, percutaneous left atrial appendage occlusion, pharmacotherapy

## Abstract

The COVID-19 pandemic has severely impacted healthcare systems worldwide. This study investigated cardiologists’ opinions on how the COVID-19 pandemic impacted clinical practice patterns in atrial fibrillation (AF). A multicenter clinician survey, including demographic and clinical questions, was administered to 300 cardiologists from 22 provinces in China, in April 2022. The survey solicited information about their treatment recommendations for AF and their perceptions of how the COVID-19 pandemic has impacted their clinical practice patterns for AF. The survey was completed by 213 cardiologists (71.0%) and included employees in tertiary hospitals (82.6%) and specialists with over 10 years of clinical cardiology practice (53.5%). Most respondents stated that there were reductions in the number of inpatients and outpatients with AF in their hospital during the pandemic. A majority of participants stated that the pandemic had impacted the treatment strategies for all types of AF, although to different extents. Compared with that during the assumed non-pandemic period in the hypothetical clinical questions, the selection of invasive interventional therapies (catheter ablation, percutaneous left atrial appendage occlusion) was significantly decreased (all *p* < 0.05) during the pandemic. There was no significant difference in the selection of non-invasive therapeutic strategies (the management of cardiovascular risk factors and concomitant diseases, pharmacotherapy for stroke prevention, heart rate control, and rhythm control) between the pandemic and non-pandemic periods (all *p* > 0.05). The COVID-19 pandemic has had a profound impact on the clinical practice patterns of AF. The selection of catheter ablation and percutaneous left atrial appendage occlusion was significantly reduced, whereas pharmacotherapy was often stated as the preferred option by participating cardiologists.

## 1. Introduction

The coronavirus disease 2019 (COVID-19) pandemic has severely disrupted medical care systems worldwide [1,2,3,4,5].The adverse influence of the pandemic on the healthcare system has impacted the prevention and treatment of COVID-19 itself, and has destabilized the relationship between clinicians and patients. In particular, the highly contagious and widespread Omicron variant has made the epidemic difficult to control and has presented challenges for the diagnosis and treatment of other common diseases [6,7].

Atrial fibrillation (AF) treatment is a comprehensive and multifaceted strategy involving heart rhythm control, heart rate control, stroke prevention therapy, interventional or surgical therapy, and the management of cardiovascular risk factors and concomitant diseases [8,9]. Several previous studies have found increased morbidity in AF patients who also have COVID-19, which affects their prognosis [10,11,12,13,14]. Zoubi M et al. demonstrated that new-onset AF is a poor prognostic sign in patients with severe COVID-19 [15]. Alkhameys S et al. performed an interrupted time series analysis of anticoagulant prescription between January 2019 and February 2021 using the English Prescribing Dataset. Although the prescription of direct-acting oral anticoagulants during the COVID-19 pandemic increased by 19%, the overall prescription of oral anticoagulants during this period was lower than expected, possibly owing to medication adherence [16]. A recently published questionnaire analysis also revealed that the COVID-19 pandemic significantly impaired the quality of life of patients awaiting AF ablation procedures [17]. Thus, COVID-19 continues to impact the management of AF and brings increasing challenges for clinicians.

However, no relevant studies have extensively investigated the impact of the COVID-19 pandemic on the clinical practice patterns of AF. In the present study, we invited 300 cardiologists from 22 provinces in China, in April 2022, to fill out a questionnaire survey investigating their perspectives on how the COVID-19 pandemic has impacted their treatment strategies for AF patients.

## 2. Methods

### 2.1. Study Design and Participants

To better understand the impact of the COVID-19 pandemic on AF treatment practices, we conducted a multicenter physician survey of 300 cardiologists in China. Research team members at the Renji Hospital, School of Medicine, Shanghai Jiaotong University, developed and distributed the survey. This survey was implemented using an online questionnaire as the primary source of data collection. The intended target population was physicians in cardiovascular departments who treated patients with AF in their routine clinical work. An initial questionnaire draft informed by the study aims was developed, pilot-tested, and revised from 1 April 2022 to 10 April 2022. A detailed questionnaire including demographic questions and clinical questions was distributed via the WeChat software to 300 cardiologists from 22 provinces in China, in April 2022. The clinician survey was administered anonymously, and all responses were submitted by 30 April 2022. This study was approved by the Institutional Ethics Committee of Renji Hospital Affiliated to Medical College of Shanghai Jiaotong University.

### 2.2. Questionnaire Design

The questionnaire, entitled “Clinician Perspectives on the Impact of the COVID-19 Pandemic on the Clinical Practice Patterns in Atrial Fibrillation”, consisted of single-choice, multiple-choice, and open-ended free-text response questions (Appendix A). Questions 1 to 5 collected the participant’s demographic information, including years of practice, subspecialty, hospital grade, and province. Questions 6 and 7 aimed to investigate the severity of the COVID-19 pandemic at the participant’s location. Questions 8 to 24 were designed to obtain the participant’s perspectives on how the COVID-19 pandemic has impacted the clinical practice patterns of AF and their treatment strategy decisions for AF patients during the pandemic. On the basis of the standard recommendations for treating AF in the latest guidelines [8,9], we explored the participant’s AF treatment practices via various clinical scenarios in questions 11 to 19. Because there is an overlap in the treatment strategies for the first diagnosed AF and other types of AF, we did not design separate questions about the first diagnosed AF. In addition, the difference between long-standing persistent AF and permanent AF mainly lies in the therapeutic attitudes of the patient and physician about the rhythm control strategy, and there is no notable difference in the clinical characteristics of these patient groups. Therefore, we combined these two types of AF when designing the questionnaire in accordance with the point of view of physicians. Clinical decisions for emergencies such as hemodynamic instability induced by AF were excluded from the clinical survey. Some rarely used AF treatments, such as AF surgery, hybrid surgical/catheter ablation procedures, atrioventricular node ablation and pacing, and surgical left atrial appendage exclusion were also not included in the questionnaire. This questionnaire defines the “previous non-pandemic period” as the year 2019 before the COVID-19 pandemic outbreak.

### 2.3. Statistical Analysis

All statistical analyses were performed using the SPSS 24.0 software (IBM Corp, Armonk, NY, USA). Continuous measures were described as mean ± standard deviation. Categorical measures were described as counts or as the number (percentage) of participants and compared by a Pearson chi-squared test. Standard *p* < 0.05 was considered statistically significant.

## 3. Results

### 3.1. Characteristics of Survey Respondents

In this multicenter survey, 300 questionnaires were distributed via WeChat. A total of 213 questionnaires were returned with a response rate of 71.0%. The characteristics of the respondents are summarized in Table 1. Among the respondents, 82.6% were employed in tertiary hospitals, 53.5% reported more than 10 years of clinical cardiology practice, 59.6% had cardiac arrhythmia as a subspecialty, and 40.8% were electrophysiologists who performed catheter ablation for arrhythmias. The locations of all participated cardiologists are summarized in Appendix A.

### 3.2. Severity of COVID-19 Pandemic at the Locations of the Participants

The participating cardiologists were from 22 provinces in China with different levels of severity of the COVID-19 pandemic. The characteristics of the pandemic are summarized in Table 2. Overall, 20.7% of participants reported more than 1000 new COVID-19 cases per day in their province during the week before responding to the survey, and 16.4% worked in a city where there were more than 1000 new COVID-19 cases per day in the week before the survey.

### 3.3. Impact of COVID-19 Pandemic on the Clinical Practice Patterns in AF

#### 3.3.1. Impact of COVID-19 Pandemic on the Numbers of AF Inpatients and Outpatients

The vast majority of respondents stated that there was an obvious reduction in the number of inpatients and outpatients with a chief complaint of AF-related symptoms, although in different proportions. Only sporadic respondents reported an increase in such patients, and a small proportion of respondents reported that there was only a slight increase or decrease (Figure 1). Similar results were seen for the number of AF patients who underwent catheter ablation therapy.

#### 3.3.2. Perception of Participating Cardiologists on the Impact of COVID-19 Pandemic on the Clinical Practice Patterns in AF

In questions 20 to 24, participants were asked: “Regarding the treatment of first diagnosed/paroxysmal/persistent/long-standing persistent/permanent AF, at what level do you think the COVID-19 pandemic has impacted you?”. The response statistics are shown in Figure 2. Less than one in five participating cardiologists stated that the pandemic had almost no impact on their treatment of first diagnosed, paroxysmal, persistent, and long-standing persistent AF. At the same time, 22.07% thought that the COVID-19 pandemic had almost no influence on their treatment of permanent AF. By contrast, the vast majority of the other participants believed that the COVID-19 pandemic had varying degrees of impact on their AF treatment practices.

#### 3.3.3. Impact of COVID-19 Pandemic on the Clinical Practice Patterns in the Paroxysmal AF

Regarding the therapeutic recommendations for patients with paroxysmal AF (Figure 3A), there were no significant differences in the management of cardiovascular risk factors and concomitant diseases or in the use of pharmacotherapy for stroke prevention, heart rate control, and rhythm control among COVID-19-positive patients, COVID-19-negative patients during the pandemic period, and patients in the non-pandemic period. However, only 13.6%, 7.5%, and 4.2% of respondents chose electrical cardioversion, catheter ablation, and percutaneous left atrial appendage occlusion, respectively, for treating COVID-19-positive patients, which were significantly lower responses than for treating COVID-19-negative patients in the pandemic period and patients in the non-pandemic period (all *p* < 0.05). Intergroup analyses revealed that the percentage of respondents recommending catheter ablation for COVID-19-negative patients with paroxysmal AF during the pandemic period was also significantly lower than in the non-pandemic period (all *p* < 0.05).

#### 3.3.4. Impact of COVID-19 Pandemic on the Clinical Practice Patterns in the Persistent AF

For patients with persistent AF, significantly fewer respondents recommended catheter ablation and percutaneous left atrial appendage occlusion for COVID-19-positive patients than for COVID-19-negative patients during the pandemic and patients in the non-pandemic period (all *p* < 0.05) (Figure 3B). Furthermore, compared with respondents who chose either of the two invasive interventional strategies for COVID-19-negative patients during the non-pandemic period, significantly fewer respondents chose either of the strategies for such patients during the pandemic period (all *p* < 0.05).

#### 3.3.5. Impact of COVID-19 Pandemic on the Clinical Practice Patterns in the Long-Standing Persistent or Permanent AF

Regarding the therapeutic recommendations for long-standing persistent or permanent AF (Figure 3C), only 4.2%, 4.2%, and 7.5% of participants chose electrical cardioversion, catheter ablation, and percutaneous left appendage occlusion, respectively, for treating COVID-19-positive patients during the pandemic period. These percentages were significantly lower than for COVID-19-negative patients during the pandemic period and patients in the non-pandemic period (all *p* < 0.05). There was no significant difference in the recommendation of electrical cardioversion, catheter ablation, and percutaneous left atrial appendage occlusion between COVID-19-negative patients during the pandemic period and non-pandemic period (all *p* > 0.05).

#### 3.3.6. Impact of COVID-19 Pandemic on the Invasive Interventional Therapies Recommended by the Participating Cardiologists for AF Patients

As shown in Figure 4, during the COVID-19 pandemic, the recommendations of invasive interventional therapies (catheter ablation and percutaneous left atrial appendage occlusion) were inversely related to the number of new COVID-19 cases in the cities of participating cardiologists (all *p* < 0.05).

## 4. Discussion

The COVID-19 pandemic has had a severe impact on healthcare systems around the world. In this multicenter physician survey, we found that the number of inpatients and outpatients with a chief complaint of AF-related symptoms and the number of AF patients who received catheter ablation therapy decreased during the pandemic period. The overwhelming majority of participating cardiologists stated that the COVID-19 pandemic had markedly affected the treatment strategies for AF patients to varying degrees. Compared with respondents who chose to use treatments for all types of AF in the period before the COVID-19 pandemic, fewer respondents chose to use invasive interventional therapies such as catheter ablation and percutaneous left atrial appendage occlusion during the pandemic period. Meanwhile, physicians were more likely to recommend pharmacotherapy for AF patients.

AF is the most common cardiac arrhythmia in adults and is associated with an increased risk of ischemic stroke, cognitive impairment, and heart failure, and it can be a considerable burden to the patient, their family, and society [18,19,20]. This study found a significant decrease in the number of inpatients and outpatients with AF treated at a hospital during the pandemic. The main reason for this phenomenon is that AF is a chronic disease, and these patients were likely worried about the potential risk of contracting COVID-19 during their hospital visit. Except for those with severe symptoms, many patients with AF chose at-home oral medication treatment. Meanwhile, some patients were also limited in traveling between regions during the pandemic, limiting their ability to attend hospital visits. Besides, COVID-19 infections among medical staff would dramatically reduce the number of available staff to provide medical services [21,22]. Although some studies identified an improvement in the physician-patient relationship during the COVID-19 pandemic, preventive strategies such as wearing face masks, face shields, and protective clothing, create barriers to effective physician-patient communication and has led to decline in trust in doctors during this challenging period [23,24]. Under such circumstances, the communication and interaction between doctors and patients are vital to ensure the comprehensive clinical management of AF. Because the COVID-19 pandemic led to a reduction in hospital visits by AF patients, it is conceivable that the ongoing pandemic could be accompanied by increases in AF-related complications and disability. In China, there are some measures to devote medical resources to the prevention and control of the COVID-19 pandemic, such as the widespread use of vaccines, nationwide free tests for COVID-19, continuous reshaping of the health emergency system, quick and effective cooperation in the joint prevention and the control of various departments, etc. [25]. The pandemic has harmed China and the global economy. That will pose a severe challenge to the health resources of countries, thereby increasing the medical burden of patients [26]. AF interventional therapies, such as catheter ablation and percutaneous left atrial appendage occlusion, are regularly carried out in general tertiary hospitals and have a unique value in the comprehensive treatment of AF. Compared with other types of medical treatment, catheter ablation can significantly prevent the recurrent of atrial arrhythmias and reduce the AF burden in any type of AF, as revealed by the recently published CABANA (Catheter Ablation versus Antiarrhythmic Drug Therapy for Atrial Fibrillation) trial [27,28]. In the AF patients enrolled in the CABANA trial who also had clinically diagnosed heart failure, catheter ablation also showed superiority for improvements in survival and quality of life when compared with medical therapy [27]. Previous studies have verified that early rhythm control therapy leads to a lower risk of cardiovascular complications in AF patients [29,30,31]. Emerging clinical evidence has also indicated that percutaneous left atrial appendage occlusion is a safe and effective therapeutic strategy for cardioembolic stroke prevention in non-valvular AF patients [32,33,34,35]. With the development of many cities in China, Shanghai has the highest rate of elderly, and the AF burden in the aging population has become substantial. Based on the data from medical insurance in the Shanghai municipal health commission database, the left atrial appendage occlusion, as an effective alternative option for AF-related stroke prevention, showed a significant increment from 0.16% in 2015 to 1.23% in 2020 [36]. Compared to other uncommon therapeutic strategies (AF surgery, hybrid surgical/catheter ablation procedures, atrioventricular node ablation and pacing, and surgical left atrial appendage exclusion), the proportion of left atrial appendage occlusion increased 7.68 times from 2015 to 2020 in Shanghai [36]. The application of left atrial appendage occlusion in AF patients who meet the indications may vary among different provinces in China. However, the present study found that the COVID-19 pandemic decreased the proportion of cardiologists who recommended catheter ablation and percutaneous left atrial appendage occlusion for AF patients (who were either positive or negative for COVID-19) during the pandemic period compared with the proportion during the non-pandemic period. According to data from the Hellenic Cardiology Society Ablation Registry, the number of ablation procedures conducted in 2019 (before the COVID-19 pandemic) and 2020 (during the COVID-19 pandemic) was reduced from 3182 cases to 2759 cases, and the number of atrial fibrillation ablation procedures was reduced by 13.8% [37]. The Spanish Catheter Ablation Registry data showed that the number of ablation procedures conducted and the success rate were both affected by the COVID-19 pandemic [38]. The quality of life of patients awaiting AF ablation has also been significantly impaired by the COVID-19 pandemic [17]. Moreover, interventional treatment during the pandemic may involve arranging a dedicated catheter room and specialized medical staff, which is a possible reason why fewer physicians chose interventional therapeutics during the pandemic than in the pre-pandemic period.

Non-invasive therapeutic strategies, such as pharmacotherapy for stroke prevention, heart rate control, rhythm control, and the management of risk factors and comorbidities, are essential cornerstones for improving long-term outcomes in patients with AF [8,9]. Handy A et al. evaluated the use of antithrombotic therapy and COVID-19 outcomes in England’s nationwide atrial fibrillation cohort. The authors found that pre-existing antithrombotic therapy was associated with lower odds of COVID-19 death in AF patients during the pandemic [39]. At the peak of COVID-19 lockdown, patients with AF-related symptoms faced various medical problems. An effective antiarrhythmic medication may be used carefully in selected AF outpatients to address these problems [40]. In the present study, there was no significant difference in the recommendations of these non-invasive strategies between the pandemic and non-pandemic periods. Although the proportion of participating physicians who recommended basic non-invasive strategies was not small, the reduction in hospital visits by AF patients during the pandemic had a potentially negative impact on the effectiveness of these treatments. Recently, the concept of Internet hospitals has become more common. Depending on the attributes of such a platform, increasing the communication between doctors and patients may be a promising approach to improve the comprehensive treatment of AF patients during the COVID-19 pandemic. Telemedicine is an important complementary strategy for the management and follow-up of some chronic diseases during the pandemic [41,42,43,44,45,46]. Previous studies have demonstrated that telemedicine could be of great value for the management of many chronic diseases during the epidemic, especially the long-term management of hypertension, chronic heart failure, and atrial fibrillation in the cardiology department during such a challenging period [41,42,43].

The current study has several limitations. Firstly, this study is a multicenter clinician survey analysis and all participated cardiologists from 22 provinces in China. Given the severity of the COVID-19 pandemic and the distribution of healthcare resources in different countries or regions, the results of this study may be biased in different regions. Secondly, the treatment of AF is complex and requires individual solutions according to guidelines’ recommendations. Thus, the hypothetical clinical scenarios based on the questionnaires may not reflect actual practice management accurately. Besides, the conclusions of this clinician survey still need to be confirmed by further clinical studies. Meanwhile, this survey mainly recruited cardiologists in hospitals that can carry out the common interventional therapeutic strategies for AF. Therefore, the higher proportions of respondents specializing in arrhythmia and working in tertiary hospitals also have potential bias.

## 5. Conclusions

In summary, the ongoing COVID-19 pandemic has had a profound impact on the clinical practice patterns of AF. During the pandemic period, the selections of catheter ablation and percutaneous left atrial appendage occlusion decreased significantly, whereas pharmacotherapy was often stated as the preferred option by participating cardiologists.

## Figures and Tables

**Figure 1 jcm-11-06469-f001:**
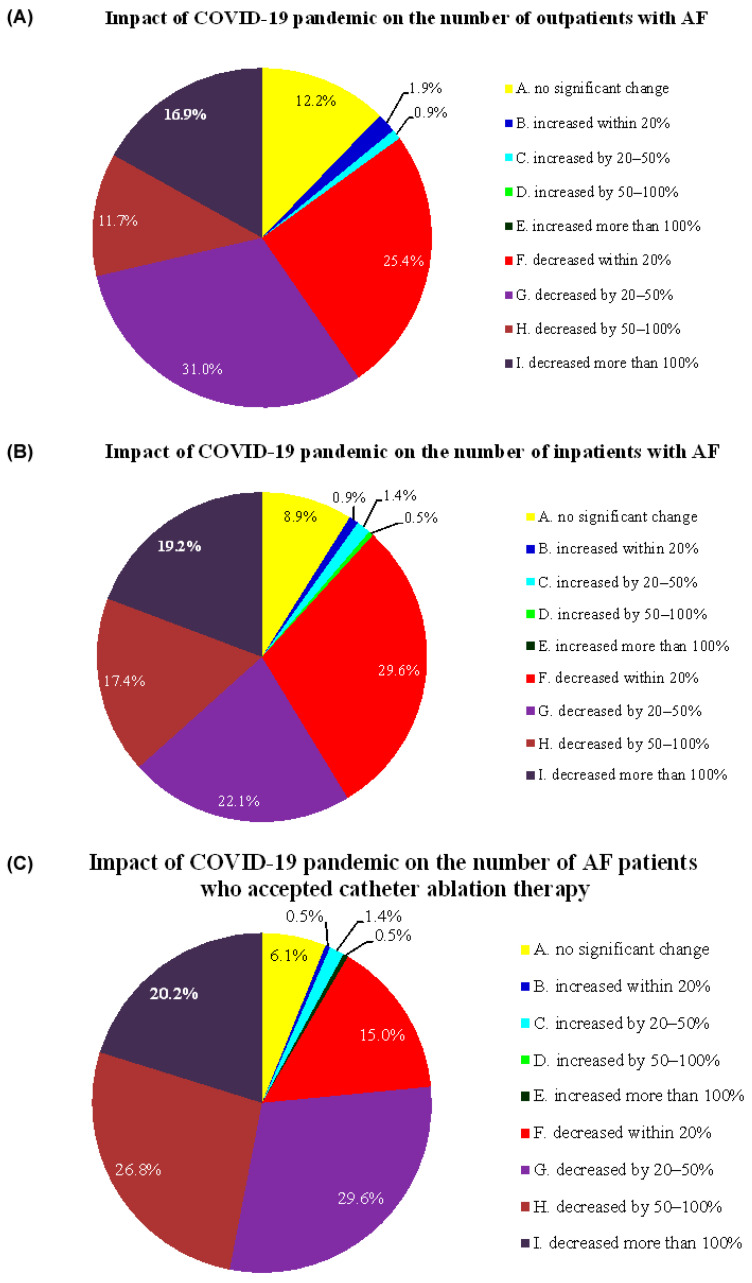
Impact of the COVID-19 pandemic on the numbers of outpatients and inpatients with AF. (**A**) Impact of the COVID-19 pandemic on the number of outpatients with AF. (**B**) Impact of the COVID-19 pandemic on the number of inpatients with AF. (**C**) Impact of the COVID-19 pandemic on the number of AF patients who underwent catheter ablation therapy.

**Figure 2 jcm-11-06469-f002:**
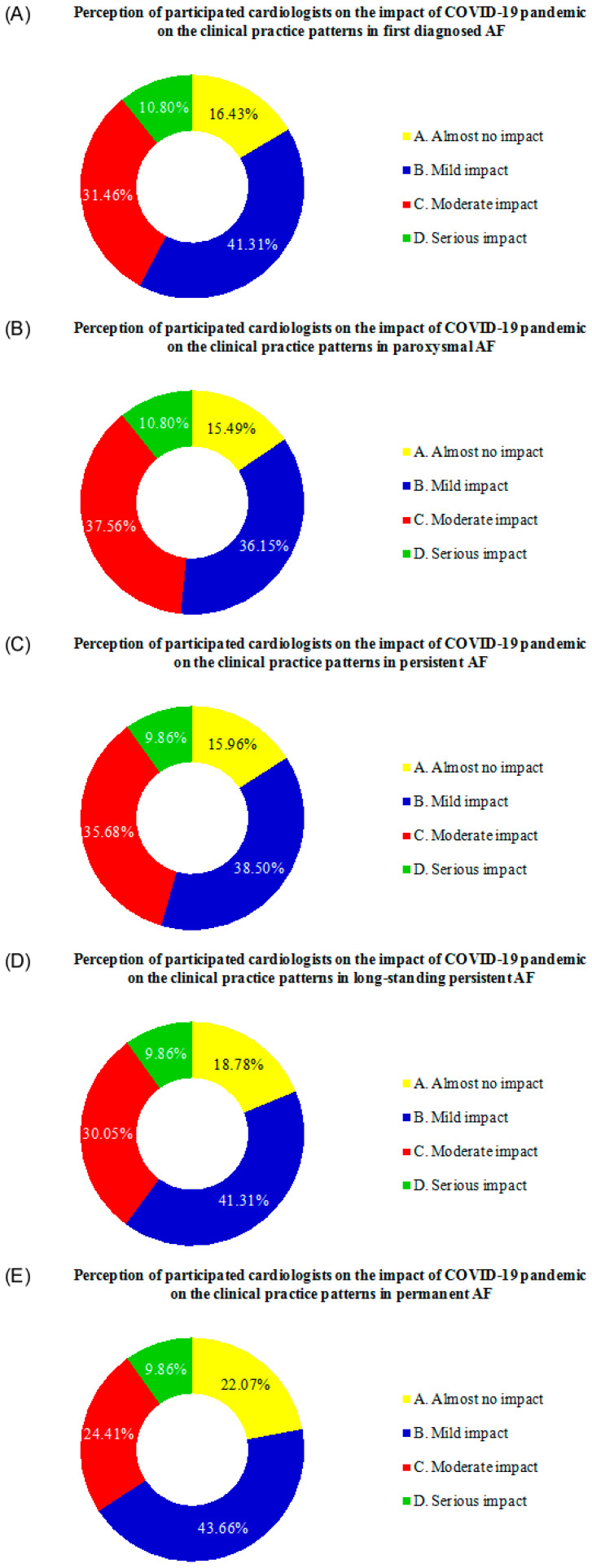
Cardiologists’ perceptions of how the COVID-19 pandemic affected AF clinical practice patterns. (**A**) Cardiologists’ perceptions on how the COVID-19 pandemic impacted the clinical practice patterns for first diagnosed AF. (**B**) Cardiologists’ perceptions on how the COVID-19 pandemic impacted the clinical practice patterns for paroxysmal AF. (**C**) Cardiologists’ perceptions on how the COVID-19 pandemic impacted the clinical practice patterns for persistent AF. (**D**) Cardiologists’ perceptions on how the COVID-19 pandemic impacted the clinical practice patterns for long-standing persistent AF. (**E**) Cardiologists’ perceptions on how the COVID-19 pandemic impacted the clinical practice patterns for permanent AF.

**Figure 3 jcm-11-06469-f003:**
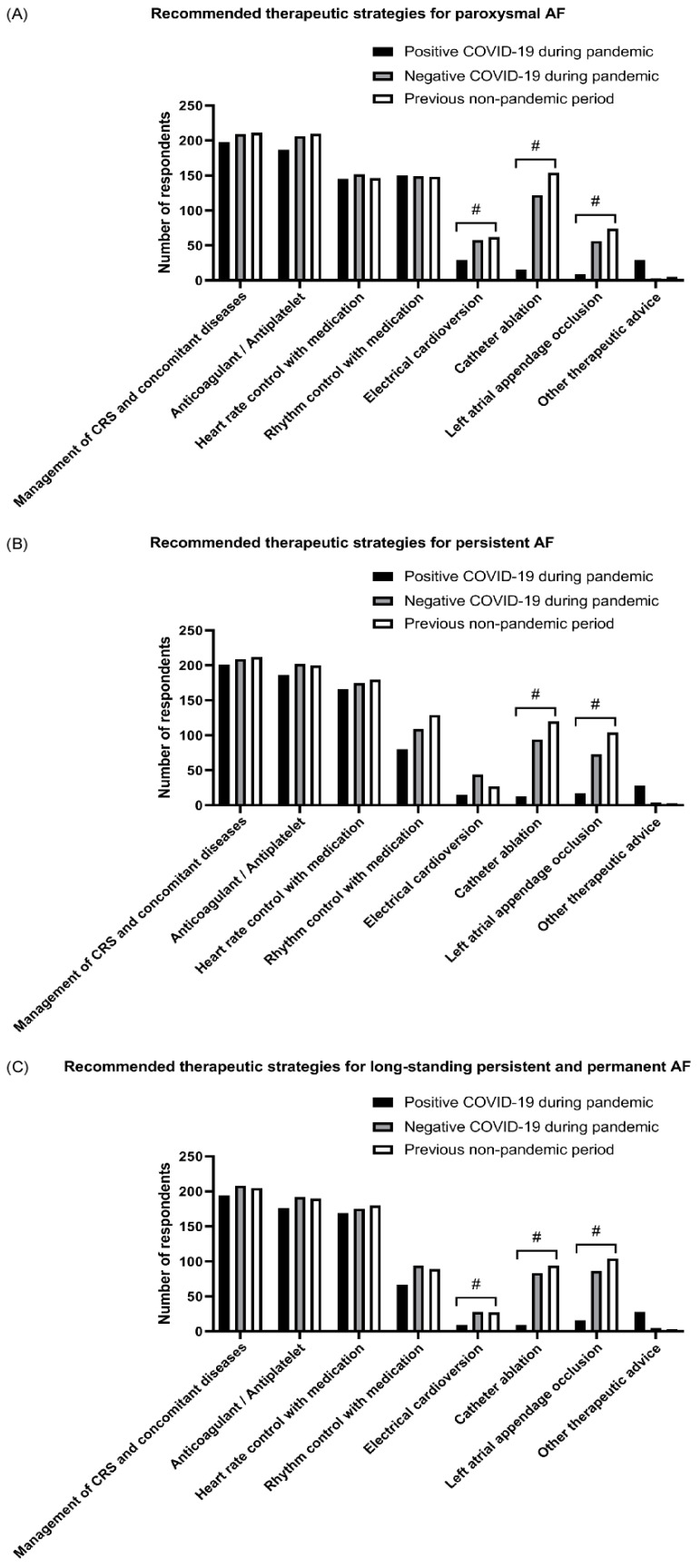
Therapeutic strategies recommended by the participating cardiologists. (**A**) Recommended therapeutic strategies for paroxysmal AF. (**B**) Recommended therapeutic strategies for persistent AF. (**C**) Recommended therapeutic strategies for long-standing persistent and permanent AF. #, *p* < 0.05.

**Figure 4 jcm-11-06469-f004:**
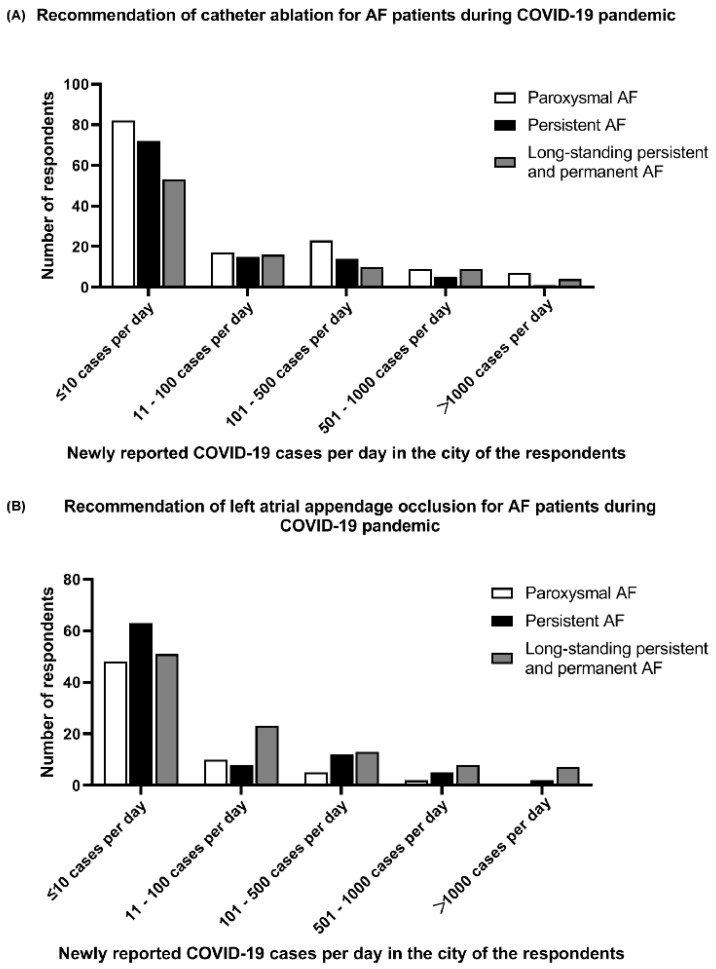
Invasive interventional therapies recommended by the participating cardiologists for AF patients during the COVID-19 pandemic. (**A**) Recommendation of catheter ablation for AF patients during COVID-19 pandemic. (**B**) Recommendation of left atrial appendage occlusion for AF patients during COVID-19 pandemic.

**Table 1 jcm-11-06469-t001:** Characteristics of survey respondents (*n =* 213).

Characteristic	*n* (%)
Years in practice (years)	
≤5	50 (23.5)
6–10	49 (23.0)
11–20	89 (41.8)
>20	25 (11.7)
Classification of employed hospital Primary general hospital Secondary general hospital	4 (1.9) 16 (7.5)
Tertiary general hospital	176 (82.6)
Cardiovascular hospital	17 (8.0)
Subspecialty (multiple choice)	
Arrhythmias	127 (59.6)
Coronary heart disease	103 (48.4)
Congenital heart disease/structural heart disease	19 (9.0)
Heart failure	64 (30.0)
Hypertension	68 (31.9)
Dyslipidemia	47 (22.1)
Critical cardiovascular diseases	36 (16.9)
Cardiovascular diseases without detailed subspecialty	50 (23.5)
Other subspecialty	10 (4.7)
Subspecialty in interventional therapy (multiple choice)	
Electrophysiology	87 (40.8)
Coronary artery intervention therapy	91 (42.7)
Cardiac device implantation	75 (35.2)
Interventional therapy for congenital heart disease	16 (7.5)
Interventional therapy for peripheral vascular diseases	3 (1.4)
Other interventional therapy	2 (0.9)
Not interventional physicians	53 (24.9)

**Table 2 jcm-11-06469-t002:** The severity of COVID-19 pandemic at the locations of the participants (*n =* 213).

Characteristic	*n* (%)
Newly reported COVID-19 cases per day in the province of the respondents	
>1000 cases per day	44 (20.66)
501–1000 cases per day	0 (0.00)
101–500 cases per day	7 (3.29)
11–100 cases per day	50 (23.47)
≤10 cases per day	112 (52.58)
Newly reported COVID-19 cases per day in the city of the respondents	
>1000 cases per day	35 (16.43)
501–1000 cases per day	3 (1.41)
101–500 cases per day	5 (2.35)
11–100 cases per day ≤10 cases per day	21 (9.86) 149 (69.95)

## Data Availability

The data used to support the findings of this study are available from the corresponding author upon reasonable request.

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
