# Peer review of "The Impact of COVID-19 Pandemic on the Clinical Practice Patterns in Atrial Fibrillation: A Multicenter Clinician Survey in China"

_jcm, 2022, doi:10.3390/jcm11216469_

Round 1

Reviewer 1 Report

An interesting study in the form of clinical survey, well organized and scientifically described. Just one observation: I believe that the following sentences (in the Discussion) "The main reason for this phenomenon is that AF is a chronic disease, and these patients were likely worried about the potential risk of contracting COVID-19 during their hospital visit. Except for those with severe symptoms, many patients with AF chose at-home oral medication treatment." should be supported by scientific references rather than be the writers' conclusion.

Author Response

Response to Reviewer's Comments:

Point 1: An interesting study in the form of clinical survey, well organized and scientifically described. Just one observation: I believe that the following sentences (in the Discussion) "The main reason for this phenomenon is that AF is a chronic disease, and these patients were likely worried about the potential risk of contracting COVID-19 during their hospital visit. Except for those with severe symptoms, many patients with AF chose at-home oral medication treatment." should be supported by scientific references rather than be the writers' conclusion.

Response 1: Thanks for your constructive comments. We modified the discussion and added the necessary references according to your suggestion. (See Page 12 Paragraph 1)

[1] Wosik J, Clowse MEB, Overton R, Adagarla B, Economou-Zavlanos N, Cavalier J, Henao R, Piccini JP, Thomas L, Pencina MJ, Pagidipati NJ. Impact of the COVID-19 pandemic on patterns of outpatient cardiovascular care. Am Heart J. 2021 Jan;231:1-5. doi: 10.1016/j.ahj.2020.10.074. Epub 2020 Nov 1. PMID: 33137309; PMCID: PMC7604084.

[2] Borrelli E, Grosso D, Vella G, Sacconi R, Querques L, Zucchiatti I, Prascina F, Bandello F, Querques G. Impact of COVID-19 on outpatient visits and intravitreal treatments in a referral retina unit: let's be ready for a plausible "rebound effect". Graefes Arch Clin Exp Ophthalmol. 2020 Dec;258(12):2655-2660. doi: 10.1007/s00417-020-04858-7. Epub 2020 Sep 22. PMID: 32960319; PMCID: PMC7505937.

Again, the authors deeply appreciate the editors and reviewers for all the efforts that you have made on this manuscript.

Sincerely yours,

Feng Hu, MD, PhD

Yan Yao, MD, PhD, FHRS

Jun Pu, MD, PhD

Reviewer 2 Report

It is well known that the Covid-19 pandemic has negatively affected every other aspect of contemporary medicine. Moreover, this study was conducted using questionneres only in China centers, which I think is a major drawback for any kind of conclusion to be made

Author Response

Response to Reviewer's Comments:

Point 1: It is well known that the Covid-19 pandemic has negatively affected every other aspect of contemporary medicine. Moreover, this study was conducted using questionneres only in China centers, which I think is a major drawback for any kind of conclusion to be made.

Response 1: Thanks for your important comments. As you kindly pointed out, this study investigated cardiologists from multiple centers in China. Given the severity of the COVID-19 pandemic and the distribution of healthcare resources in different countries or regions, the results of this study may be biased in different regions. Frankly speaking, it is challenging for us to conduct global-scale research in the current state of epidemic ravaging and global instability. Considering that the pandemic is still raging worldwide, we hope this study can provide a modest hint for the healthcare system and clinical physicians. We also added this point as a limitation in the revised manuscript. (See Page 13, Paragraph 3).

Again, the authors deeply appreciate the editors and reviewers for all the efforts that you have made on this manuscript.

Sincerely yours,

Feng Hu, MD, PhD

Yan Yao, MD, PhD, FHRS

Jun Pu, MD, PhD

Reviewer 3 Report

The authors performed a multicenter survey in order to investigate how COVID-19 pandemic impacted on treatment strategies of patients affected by atrial fibrillation. They submitted anonymously an online  questionnaire to 300 cardiologists  from 22 Chinese provinces in April 2022. A total of 231 questionnaire ( response rate of 71%) collected demographic information, incidence of COVID-19 at the participant’s location, clinical patterns of AF and treatment strategies adopted by physicians. Exclusion criteria were: AF-related hemodynamic instability and not common therapeutic advices (such as, AF surgery/hybrid/catether ablation, ablate and pacing…). 

The authors found that: 

-       the number of inpatients and outpatients with a chief-complaint of AF-related symptoms was decreased during the pandemic;

-       invasive strategies (catheter ablation, percutaneous left atrial appendage occlusion) were  significantly decreased during the pandemic;

-       pharmacological strategies for stroke prevention, heart rate control and rhythm control were preferred options by participants either in pandemic and non-pandemic period.

Overall comment:

The topic could be interesting, however the paper has little clinical impact ad there are some important concerns. During the pandemic worldwide healthcare systems were focused on life-saving procedures and patients were concerned about contracting Sars-Cov2 infection during hospitalization for deferrable procedures such as catheter ablation (CA). In fact, according to European guidelines’ recommendations in the context of COVID19 (23 April 2021), catheter ablation was not an urgent therapy, except for tachycardiomyopathy or syncope. 

Detailed comment below: 

1.     Contrary to what the authors said in the discussion section, CABANA TRIAL showed clear superiority of CA for prevention of recurrent atrial arrhythmias and reduction of AF burden but it failed to demonstrate a superiority of CA compared to medical therapy in the prevention of  death, disabling stroke, serious bleeding or cardiac arrest due to its important limitations (single blind trial,  intention-to-treat randomization and different approaches in control group).

2.     Regarding exclusion criteria, it should be better adding left appendage closure among not common therapeutic advices. In fact, the left appendage closure is indicated in a selected cohort of patients such as those with high risk of bleeding, with contraindication to OAC or inefficient anticoagulation (“stroke in warfarin”). 

3.     The authors specified when responses were submitted but they did not specify the time period the questionnaires are referred to. In addition, they wrote “pre-pandemic period” without giving precise time frame. 

4.     Supplement 1 with the questionnaire is not available.

5.     Baseline characteristics of patients to which respondants referred in questionnaires are unknown. It could be necessary in order to compare pre-pandemic and pandemic population.

6.     Population of the respondents is not homogeneous. In fact, cardiologists subspecialized in arrhythmias were over-represented (59.6%), so they might have performed CA less strictly during pre-pandemic period. In addition, among the respondents, 82.6% were employed in tertiary hospitals where invasive procedures are more accessible compared to primary or secondary centers. 

7.     In Table 2, the authors gave the rates of incidence of new COVID19 cases at the locations of the participants. However, in the survey, the authors did not specify if tertiary hospitals (in which invasive procedures were regularly carried out) were located in provinces or cities with higher incidence rates of SARS-COV2 infections, causing a decreased number of CAs or LAAOs during the pandemic. 

8.     In the paragraph 3.2, it is not clear why new COVID19 cases are referred only to the week before responding to the survey and not to the whole pandemic period.

9.     In the paragraph 3.3.1 the majority of respondents reported a decreased number of symptomatic cases of atrial fibrillation during the pandemic. Any explanation was given by the authors in the discussion of the survey.

10.  In the Figure 3, we read “Other therapeutic advice”. What are they? Previously, the authors stated that AF surgery,  hybrid surgical/cathether ablation procedures, atrioventricular node ablation and pacing and surgical LAAO were not included in the questionnaire.

11.  In the paragraph 2.2 about “Questionnaire design”, it is not clear what “latest guidelines” the authors were referring to.

12.  Conclusion section is missing 

13.  Graphics quality need to be reviewed. In particular, in the figure 1, graphs’ percentages are not so easy to read.

In conclusion, the paper is not original and has little clinical impact. For example, it would be more interesting to investigate the incidence of stroke related to misdiagnosed AF during the pandemic or how changed the clinical practice of AF in pre- and post-vaccination period.

Author Response

Response to Reviewer's Comments

Point 1: Contrary to what the authors said in the discussion section, CABANA TRIAL showed clear superiority of CA for prevention of recurrent atrial arrhythmias and reduction of AF burden but it failed to demonstrate a superiority of CA compared to medical therapy in the prevention of death, disabling stroke, serious bleeding or cardiac arrest due to its important limitations (single blind trial, intention-to-treat randomization and different approaches in control group).

Response 1: Sincerely thanks for your important comments. As you kindly pointed out, CABANA TRIAL failed to demonstrate the superiority of CA compared to medical therapy in preventing death, disabling stroke, severe bleeding, or cardiac arrest due to its limitations. In the AF patients enrolled in the CABANA trial who also had clinically diagnosed heart failure, catheter ablation showed superiority for improved survival and quality of life compared to medical therapy.1 We apologized for not expressing these points very clearly and modified the Discussion according to your suggestion. (See Page 12, Paragraph 2)

[1]  Packer DL, Piccini JP, Monahan KH, Al-Khalidi HR, Silverstein AP, Noseworthy PA, et al. Ablation Versus Drug Therapy for Atrial Fibrillation in Heart Failure: Results From the CABANA Trial. Circulation.(2021) 143:1377-1390.doi: 10.1161/CIRCULATIONAHA.120.050991.

Point 2: Regarding exclusion criteria, it should be better adding left appendage closure among not common therapeutic advices. In fact, the left appendage closure is indicated in a selected cohort of patients such as those with high risk of bleeding, with contraindication to OAC or inefficient anticoagulation (“stroke in warfarin”).

Response 2: Thanks for your constructive comments. Based on the development of many cities in China, Shanghai has the highest rate of elderly, where the AF burden has become substantial. Based on the data from medical insurance in the Shanghai Municipal Health Commission database, the left atrial appendage closure (LAAC), as an effective alternative option for AF-related stroke prevention, showed a surge from 0.16% in 2015 to 1.23% in 2020.2 Compared to other uncommon therapeutic strategies (AF surgery,  hybrid surgical/catheter ablation procedures, atrioventricular node ablation and pacing, and surgical LAAC), the proportion of LAAC increased 7.68 times from 2015 to 2020 in Shanghai. Thus we set LAAC as an option during this survey research. Frankly speaking, there may be differences in medical investment among different provinces in China, and the proportion of LAAC in AF patients who meet the indications may not be the same. Sincerely thank you.

LAAC application in AF patients between 2015 and 2020 in Shanghai [2]

[2]  Chen M, Li C, Liao P, Cui X, Tian W, Wang Q, Sun J, Yang M, Luo L, Wu H, Li YG. Epidemiology, management, and outcomes of atrial fibrillation among 30 million citizens in Shanghai, China from 2015 to 2020: A medical insurance database study. Lancet Reg Health West Pac. 2022 May 3;23:100470. doi:10.1016/j.lanwpc.2022.100470. PMID: 35542895; PMCID: PMC9079299.

Point 3: The authors specified when responses were submitted but they did not specify the time period the questionnaires are referred to. In addition, they wrote “pre-pandemic period” without giving precise time frame.

Response 3: According to your constructive comments, we added this survey's detailed development and deployment in the revised manuscript (See Page 2, Paragraph 4). An initial questionnaire draft informed by the study aims was developed, pilot-tested, and revised from April 1, 2022, to April 10, 2022. A detailed questionnaire including demographic questions and clinical questions was distributed via WeChat software to 300 cardiologists from 22 provinces in China. The clinician survey was administered anonymously, and all responses were submitted by April 30, 2022. In this questionnaire, the “previous non-pandemic period” was defined as the year of 2019 before the COVID-19 pandemic outbreak (See Page 3, Paragraph 1 and Supplement 1).

Point 4: Supplement 1 with the questionnaire is not available.

Response 4: Sincerely thanks for your important comments. We have uploaded Supplement 1 as an attachment.

Point 5: Baseline characteristics of patients to which respondants referred in questionnaires are unknown. It could be necessary in order to compare pre-pandemic and pandemic population.

Response 5: Sincerely thanks for your comments. The intended target population was physicians in cardiovascular departments who treated patients with AF in their routine clinical work. In the questionnaire, we set the type of AF patients via various clinical scenarios in questions 11 to 19. (See Supplement 1)

Point 6: Population of the respondents is not homogeneous. In fact, cardiologists subspecialized in arrhythmias were over-represented (59.6%), so they might have performed CA less strictly during pre-pandemic period. In addition, among the respondents, 82.6% were employed in tertiary hospitals where invasive procedures are more accessible compared to primary or secondary centers.

Response 6: Sincerely thanks for your important comments. Evaluating the pandemic's impact on the interventional treatment of atrial fibrillation is an important objective of this study. In the original intention of the research design, we mainly recruited cardiologists in medical institutions which could carry out the common interventional therapeutic strategies for AF patients. Frankly, the higher proportions of respondents specializing in arrhythmia and working in tertiary hospitals also have potential bias. We also added this point as a limitation in the revised manuscript. (See Page 13, Paragraph 3)

Point 7: In Table 2, the authors gave the rates of incidence of new COVID19 cases at the locations of the participants. However, in the survey, the authors did not specify if tertiary hospitals (in which invasive procedures were regularly carried out) were located in provinces or cities with higher incidence rates of SARS-COV2 infections, causing a decreased number of CAs or LAAOs during the pandemic.

Response 7: Enlightened by your constructive comment, we respectively analyzed the data and added a new figure 4 to illustrate the relationship between the recommendations of catheter ablation and LAAO among the respondents with different new COVID-19 cases in their cities during the COVID-19 pandemic. As shown in Figure 4, during the COVID-19 pandemic, the recommendations of invasive interventional therapies (catheter ablation and percutaneous left atrial appendage occlusion) were inversely related to the number of new COVID-19 cases in the cities of participating cardiologists (all P<0.05).

Figure 4. Invasive interventional therapies recommended by the participating cardiologists for AF patients during the COVID-19 pandemic. (A) Recommendation of catheter ablation for AF patients during COVID-19 pandemic. (B) Recommendation of left atrial appendage occlusion for AF patients during COVID-19 pandemic.

Point 8: In the paragraph 3.2, it is not clear why new COVID-19 cases are referred only to the week before responding to the survey and not to the whole pandemic period.

Response 8: Thanks for your important comments. Due to the anti-epidemic strategies being challenging in China, the new COVID-19 cases in the provinces and cities are changing rapidly. In this study, we aimed to explore whether the COVID-19 pandemic influences the choice of therapeutic strategies for AF patients by participating doctors, so the new COVID-19 cases are referred only to the week before responding to the survey. In addition, as a survey study, if the questionnaire requires respondents to calculate the whole number of COVID-19 cases throughout the epidemic, it may increase the difficulty of data collection and may further reduce the response rate of this survey.

Point 9: In the paragraph 3.3.1 the majority of respondents reported a decreased number of symptomatic cases of atrial fibrillation during the pandemic. Any explanation was given by the authors in the discussion of the survey.

Response 9: Several reasons may affect the number of hospital visits of patients with AF.3-4 The main reason for this phenomenon is that AF is a chronic disease, and these patients were likely worried about the potential risk of contracting COVID-19 during their hospital visit. Except for those with severe symptoms, many patients with AF chose at-home oral medication treatment. Meanwhile, some patients were also limited in traveling between regions during the pandemic, limiting their ability to attend hospital visits. Besides, COVID-19 infections among medical staff would dramatically reduce the number of available staff to provide medical services. We modified the discussion and added the necessary references according to your suggestion. (See Page 12, Paragraph 1)

[3]  Wosik J, Clowse MEB, Overton R, Adagarla B, Economou-Zavlanos N, Cavalier J, Henao R, Piccini JP, Thomas L, Pencina MJ, Pagidipati NJ. Impact of the COVID-19 pandemic on patterns of outpatient cardiovascular care. Am Heart J. 2021 Jan;231:1-5. doi: 10.1016/j.ahj.2020.10.074. Epub 2020 Nov 1. PMID: 33137309; PMCID: PMC7604084.

[4]  Borrelli E, Grosso D, Vella G, Sacconi R, Querques L, Zucchiatti I, Prascina F, Bandello F, Querques G. Impact of COVID-19 on outpatient visits and intravitreal treatments in a referral retina unit: let's be ready for a plausible "rebound effect". Graefes Arch Clin Exp Ophthalmol. 2020 Dec;258(12):2655-2660. doi: 10.1007/s00417-020-04858-7. Epub 2020 Sep 22. PMID: 32960319; PMCID: PMC7505937.

Point 10: In the Figure 3, we read “Other therapeutic advice”. What are they? Previously, the authors stated that AF surgery,  hybrid surgical/cathether ablation procedures, atrioventricular node ablation and pacing and surgical LAAO were not included in the questionnaire.

Response 10: In the questionnaire, “Other therapeutic advice” is an open fill-in-the-blank option for questions 11-19. The proportion of participating cardiologists who select these options is tiny. The answers to these options were scattered and different (individualized medicine, individualized ablation, combined therapy of catheter ablation and left atrial appendage closure, et al ……), thus is really difficult to perform the statistical analysis for the detailed each answer. We just gather these answers together as “other therapeutic advice” in the result.

Point 11: In the paragraph 2.2 about “Questionnaire design”, it is not clear what “latest guidelines” the authors were referring to.

Response 11: We added the necessary references in the revised manuscript according to your critical comments. (See Page 3, Paragraph 1)

Point 12: Conclusion section is missing. 

Response 12: We added the Conclusion section in the revised manuscript.

Point 13: Graphics quality need to be reviewed. In particular, in the figure 1, graphs’ percentages are not so easy to read.

Response 13: Thanks for your critical comments. We revised Figure 1 according to your suggestion. (See new Figure 1)

In this multicenter physician survey, we found that the COVID-19 pandemic has profoundly impacted the clinical practice patterns of AF. We totally agree with you that during the COVID-19 pandemic, our healthcare systems should focus on life-saving procedures for patients with Sars-Cov2 infection. In the questionnaire, we designed various clinical scenarios in questions 11 to 19 to explore the cardiologists’ therapeutic tendency for AF patients with positive or negative COVID-19 infection during the pandemic and AF patients in the previous non-pandemic period. The questions and options are relatively objective, and the study results may reflect some existing problems among the participating physicians. Frankly speaking, it is challenging for us to conduct global-scale research in the current state of epidemic ravaging and global instability. Considering that the pandemic is still raging worldwide, we hope this study can provide a modest hint for the healthcare system and clinical physicians.

Again, the authors deeply appreciate the editors and reviewers for all the efforts that you have made on this manuscript.

Sincerely yours,

Feng Hu, MD, PhD

Yan Yao, MD, PhD, FHRS

Jun Pu, MD, PhD

Reviewer 4 Report

The question that moves your research is valid and current, addressing a global impact issue such as the management of a chronic disease in a complex scenario from a health, as well as a social point of view, such as the COVID-19 pandemic. It would be useful to provide the reader with a further key to the interpretation of the data exposed, explaining the structure of the local health organization, both in the pre-pandemic era and as any changes in access and provision of care with the advent of COVID-19 (for example, costs and methods of providing visits, reorganization of health structures).

Some aspects should be improved:

1)      Please discuss why was the survey only completed by 71% of the cardiologists interviewed? Would it be possible to try to decline the low participation, if in terms of scarce time available or low interest in the topic dealt with?

2)      Based on these considerations, what could have affected the management of patients suffering from atrial fibrillation respectively the fear of accessing health facilities (with the relative fear in terms of exposure to contagion) and how much the diversion of health resources devolved conspicuously towards the management of the COVID-19 emergency? And any economic difficulties of patients in accessing care (considering the working conditions made precarious by the contextual social upheaval)?

As you mentioned, the survey highlights how even the difficulty of the pandemic scenario, by altering the usual doctor-patient relationship, can at the same time implement tele-medicine as an increasingly important tool in guaranteeing an ongoing relationship with fragile patients suffering from chronic pathologies. Contact in person can certainly not be replaced by telephone or video contact, but if the latter is assisted by a valid local medical-nursing network it can lead to intercepting problems (for example, poor heart rate control) in terms of cardiological acuity and related repercussions in terms of prognosis, quality of life and health costs.

3)      How is the health organization through tele-medicine in your region? How widespread and applied is it? Why not investigate the application and the degree of possible adhesion by cardiologists in the setting of your survey? Please discuss the importance of telemedicine during COVID-19 pandemic era and future perspectives based on the lesson learned during that period (please see . J Clin Med. 2022 May 16;11(10):2790. doi: 10.3390/jcm11102790 and Circ J. 2020 Sep 25;84(10):1679-1685. doi: 10.1253/circj.CJ-20-0566. )

Author Response

Response to Reviewer's Comments:

Point 1: Please discuss why was the survey only completed by 71% of the cardiologists interviewed? Would it be possible to try to decline the low participation, if in terms of scarce time available or low interest in the topic dealt with?

Response 1: Sincerely thanks for your important comments. As you kindly pointed out, only 71% of the interviewed cardiologists completed the survey, which may potentially influence the results. There may be several reasons for the relatively low participation in the study. On the one hand, during the COVID-19 pandemic, almost all hospitals in China participated in the fight against COVID-19, and the workload of medical staff has become extremely difficult. On the other hand, although all invited doctors are cardiovascular specialists, 59.6% had cardiac arrhythmia as a subspecialty, and some other invited cardiologists had other subspecialties. We described the 213 participated cardiologists in the revised Method in detail and added their provinces and proportions in the revised manuscript (See Page 3 and Supplemental Table 1). They are distributed in different medical institutions in 22 provinces of China, and have certain representation (See Supplemental Table 1).

Point 2: Based on these considerations, what could have affected the management of patients suffering from atrial fibrillation respectively the fear of accessing health facilities (with the relative fear in terms of exposure to contagion) and how much the diversion of health resources devolved conspicuously towards the management of the COVID-19 emergency? And any economic difficulties of patients in accessing care (considering the working conditions made precarious by the contextual social upheaval)?

As you mentioned, the survey highlights how even the difficulty of the pandemic scenario, by altering the usual doctor-patient relationship, can at the same time implement tele-medicine as an increasingly important tool in guaranteeing an ongoing relationship with fragile patients suffering from chronic pathologies. Contact in person can certainly not be replaced by telephone or video contact, but if the latter is assisted by a valid local medical-nursing network it can lead to intercepting problems (for example, poor heart rate control) in terms of cardiological acuity and related repercussions in terms of prognosis, quality of life and health costs.

Response 2: Thanks for your constructive comments. Several reasons may affect the number of hospital visits of patients with AF.1-2 The main reason for this phenomenon is that AF is a chronic disease, and these patients were likely worried about the potential risk of contracting COVID-19 during their hospital visit. Except for those with severe symptoms, many patients with AF chose at-home oral medication treatment. Meanwhile, some patients were also limited in traveling between regions during the pandemic, limiting their ability to attend hospital visits. Besides, COVID-19 infections among medical staff would dramatically reduce the number of available staff to provide medical services. We modified the discussion and added the necessary references according to your suggestion. (See Page 12, Paragraph 1)

In China, there are some measures to devote medical resources to the prevention and control of the COVID-19 pandemic, such as widespread use of vaccines, nationwide free tests for COVID-19, continuous reshaping of the health emergency system, quick and effective cooperation in the joint prevention and control of various departments, etc.3 The pandemic has harmed China and the global economy. That will pose a severe challenge to the health resources of countries, thereby increasing the medical burden of patients.4

This study was designed as a survey of clinical cardiologists to investigate how the COVID-19 pandemic has impacted their treatment strategies for AF patients. Thus we did not include the exploration of the social medical resources and the economic burden of patients during the study design. In addition, as a survey study, if the questionnaire requires respondents to calculate the social medical resources and the economic burden of patients, it may increase the difficulty of data collection and may further reduce the response rate of this survey. We have studied relevant literatures and added this section to the latest discussion section. (See Page 12, Paragraph 1)

As you kindly pointed out, telemedicine is an important complementary strategy for managing and following chronic diseases during the epidemic. Previous studies have demonstrated that telemedicine could be of great value for the management of many chronic diseases during the epidemic,5-10 especially the long-term management of hypertension, chronic heart failure, and atrial fibrillation in the cardiology department, and this is also our future direction to improve the medical service level during such a challenging period. We also added these points in the revised Discussion (See Page 13, Paragraph 2).

[1]  Wosik J, Clowse MEB, Overton R, Adagarla B, Economou-Zavlanos N, Cavalier J, Henao R, Piccini JP, Thomas L, Pencina MJ, Pagidipati NJ. Impact of the COVID-19 pandemic on patterns of outpatient cardiovascular care. Am Heart J. 2021 Jan;231:1-5. doi: 10.1016/j.ahj.2020.10.074. Epub 2020 Nov 1. PMID: 33137309; PMCID: PMC7604084.

[2]  Borrelli E, Grosso D, Vella G, Sacconi R, Querques L, Zucchiatti I, Prascina F, Bandello F, Querques G. Impact of COVID-19 on outpatient visits and intravitreal treatments in a referral retina unit: let's be ready for a plausible "rebound effect". Graefes Arch Clin Exp Ophthalmol. 2020 Dec;258(12):2655-2660. doi: 10.1007/s00417-020-04858-7. Epub 2020 Sep 22. PMID: 32960319; PMCID: PMC7505937.

[3]  Wang J, Wang Z. Strengths, Weaknesses, Opportunities and Threats (SWOT) Analysis of China's Prevention and Control Strategy for the COVID-19 Epidemic. Int J Environ Res Public Health. 2020 Mar 26;17(7):2235. doi: 10.3390/ijerph17072235. PMID: 32225019; PMCID: PMC7178153.

[4]  Shen J, Shum WY, Cheong TS, Wang L. COVID-19 and Regional Income Inequality in China. Front Public Health. 2021 May 11;9:687152. doi: 10.3389/fpubh.2021.687152. PMID: 34046393; PMCID: PMC8144473.

[5]  Severino P, D'Amato A, Prosperi S, Magnocavallo M, Maraone A, Notari C, Papisca I, Mancone M, Fedele F. Clinical Support through Telemedicine in Heart Failure Outpatients during the COVID-19 Pandemic Period: Results of a 12-Months Follow Up. J Clin Med. 2022 May 16;11(10):2790. doi: 10.3390/jcm11102790. PMID:35628916; PMCID: PMC9147859.

[6]  Hu YF, Cheng WH, Hung Y, Lin WY, Chao TF, Liao JN, Lin YJ, Lin WS, Chen YJ, Chen SA. Management of Atrial Fibrillation in COVID-19 Pandemic. Circ J. 2020 Sep 25;84(10):1679-1685. doi: 10.1253/circj.CJ-20-0566. Epub 2020 Sep 9. PMID: 32908073.

[7]  Colbert GB, Venegas-Vera AV, Lerma EV. Utility of telemedicine in the COVID-19 era. Rev Cardiovasc Med. 2020 Dec 30;21(4):583-587. doi: 10.31083/j.rcm.2020.04.188. PMID: 33388003.

[8]  Gareev I, Gallyametdinov A, Beylerli O, Valitov E, Alyshov A, Pavlov V, Izmailov A, Zhao S. The opportunities and challenges of telemedicine during COVID-19 pandemic. Front Biosci (Elite Ed). 2021 Dec 20;13(2):291-298. doi: 10.52586/E885. PMID: 34937315.

[9]  Bokolo AJ. Exploring the adoption of telemedicine and virtual software for care of outpatients during and after COVID-19 pandemic. Ir J Med Sci. 2021 Feb;190(1):1-10. doi: 10.1007/s11845-020-02299-z. Epub 2020 Jul 8. PMID: 32642981; PMCID: PMC7340859.

[10]  Ohannessian R, Duong TA, Odone A. Global Telemedicine Implementation and Integration Within Health Systems to Fight the COVID-19 Pandemic: A Call to Action. JMIR Public Health Surveill. 2020 Apr 2;6(2):e18810. doi: 10.2196/18810. PMID: 32238336; PMCID: PMC7124951.

Point 3: How is the health organization through tele-medicine in your region? How widespread and applied is it? Why not investigate the application and the degree of possible adhesion by cardiologists in the setting of your survey? Please discuss the importance of telemedicine during COVID-19 pandemic era and future perspectives based on the lesson learned during that period (please see . J Clin Med. 2022 May 16;11(10):2790. doi: 10.3390/jcm11102790 and Circ J. 2020 Sep 25;84(10):1679-1685. doi: 10.1253/circj.CJ-20-0566. )

Response 3: Thanks for your constructive comments. Renji Hospital has built a telemedicine software system supported by mobile phones (see the following Picture), but it is still mainly used for the consultation of some simple diseases and the dispensing of medications for patients with chronic diseases, and lacks systematic management of specific illnesses. This is the status quo of a tertiary hospital in Shanghai, which may be more inferior in other regions of China. Therefore, we did not design a special questionnaire in this regard. As you said, telemedicine could be of great value for the management of many chronic diseases during the epidemic, especially the long-term management of hypertension, chronic heart failure, and atrial fibrillation in the cardiology department, 5-10 and this is also our future direction to improve the medical service level during such a challenging period. Sincerely thanks for the references you recommended, we have learned the relevant literatures carefully and added these points in the revised Discussion (See Page 13, Paragraph 2).

[5]  Severino P, D'Amato A, Prosperi S, Magnocavallo M, Maraone A, Notari C, Papisca I, Mancone M, Fedele F. Clinical Support through Telemedicine in Heart Failure Outpatients during the COVID-19 Pandemic Period: Results of a 12-Months Follow Up. J Clin Med. 2022 May 16;11(10):2790. doi: 10.3390/jcm11102790. PMID:35628916; PMCID: PMC9147859.

[6]  Hu YF, Cheng WH, Hung Y, Lin WY, Chao TF, Liao JN, Lin YJ, Lin WS, Chen YJ, Chen SA. Management of Atrial Fibrillation in COVID-19 Pandemic. Circ J. 2020 Sep 25;84(10):1679-1685. doi: 10.1253/circj.CJ-20-0566. Epub 2020 Sep 9. PMID: 32908073.

[7]  Colbert GB, Venegas-Vera AV, Lerma EV. Utility of telemedicine in the COVID-19 era. Rev Cardiovasc Med. 2020 Dec 30;21(4):583-587. doi: 10.31083/j.rcm.2020.04.188. PMID: 33388003.

[8]  Gareev I, Gallyametdinov A, Beylerli O, Valitov E, Alyshov A, Pavlov V, Izmailov A, Zhao S. The opportunities and challenges of telemedicine during COVID-19 pandemic. Front Biosci (Elite Ed). 2021 Dec 20;13(2):291-298. doi: 10.52586/E885. PMID: 34937315.

[9]  Bokolo AJ. Exploring the adoption of telemedicine and virtual software for care of outpatients during and after COVID-19 pandemic. Ir J Med Sci. 2021 Feb;190(1):1-10. doi: 10.1007/s11845-020-02299-z. Epub 2020 Jul 8. PMID: 32642981; PMCID: PMC7340859.

[10]  Ohannessian R, Duong TA, Odone A. Global Telemedicine Implementation and Integration Within Health Systems to Fight the COVID-19 Pandemic: A Call to Action. JMIR Public Health Surveill. 2020 Apr 2;6(2):e18810. doi: 10.2196/18810. PMID: 32238336; PMCID: PMC7124951.

Medical software system supported by mobile phones in Renji Hospital

Again, the authors deeply appreciate the editors and reviewers for all the efforts that you have made on this manuscript.

Sincerely yours,

Feng Hu, MD, PhD

Yan Yao, MD, PhD, FHRS

Jun Pu, MD, PhD

Round 2

Reviewer 2 Report

The revised version has covered the issues mentioned.

Author Response

Thank you very much for the excellent and professional revision of our manuscript. 

We have checked the manuscript by a native English editor during submission.

Sincerely yours,
Feng Hu, MD, PhD
Yan Yao, MD, PhD, FHRS
Jun Pu, MD, PhD

Reviewer 3 Report

Manuscript number: JCM_2022_1908443

Title:  The impact of COVID-19 pandemic on the clinical practice patterns in atrial fibrillation: A multicenter clinician survey in China

I read the revised version of “The impact of COVID-19 pandemic on the clinical practice patterns in atrial fibrillation: A multicenter clinician survey in China” written by Feng Hu et alii in which the authors concluded that COVID19 pandemic had a profound impact on the clinical management of in- and out-patients affected by atrial fibrillation. Pharmacotherapy was preferred to catheter ablation (CA) and percutaneous left atrial appendage closure (LAAO) by participating cardiologists during pandemic period. There results seems to support what the most important International Guidelines stated about the management of atrial fibrillation in the context of COVID19.

The authors have improved the study design specifying better the time frame and adding the text of the questionnaire, which can be now consulted by readers.

Results are clearer, especially in section 3.3.6 in which the authors have illustrated the relationship between the recommendations of CA and LAAO with different COVID 19 severity at the locations of the participants. In addition, graphs and tables are more easily accessible.

Discussion is more comprehensive because the authors have explained better why the number of symptomatic AF patients is decreased during the pandemic, they have commented more appropriately the results of CABANA trial and, even more so, they have correctly highlighted the potential bias represented by the higher proportions of recruited sub-specialized cardiologists in arrhythmia.

They have also added the conclusion section that was previously missed.

I sincerely appreciate the “Response 2” on left appendage closure as alternative option for AF-related stroke prevention in many cities in China but I think it could be added in the discussion section.

In conclusion, even if this paper has little clinical impact, it could be published after this last minor revision.

Author Response

Thank you very much for the excellent and professional revision of our manuscript.

According to your constructive comments, we added the discussion about the left atrial appendage closure as alternative option for AF-related stroke prevention in China (See Page 12, Paragraph 2).

Sincerely yours,

Feng Hu, MD, PhD

Yan Yao, MD, PhD, FHRS

Jun Pu, MD, PhD